# Restored Intensities from Customized Crops of NMR Experiments (RICC-NMR) to Gain Better Insight on Chemometrics of Sicilian and Sardinian Extra Virgin Olive Oils

**DOI:** 10.3390/foods14101807

**Published:** 2025-05-19

**Authors:** Nicola Culeddu, Archimede Rotondo, Federico Nastasi, Giovanni Bartolomeo, Pierfrancesco Deiana, Mario Santona, Petros A. Tarantilis, Giuseppe Pellicane, Giovanna Loredana La Torre

**Affiliations:** 1CNR—Istituto di Chimica Biomolecolare (ICB), Traversa La Crucca 3, Loc. Baldinca, Li Punti, 07040 Sassari, Italy; nicola.culeddu@cnr.it; 2Department of Biomedical and Dental Sciences and Morpho-Functional Imaging (BIOMORF), University of Messina, 98168 Messina, Italy; gbartolomeo@unime.it (G.B.); gpellicane@unime.it (G.P.); 3Dipartimento di Scienze Chimiche, Biologiche, Farmaceutiche ed Ambientali (CHIBIOFARAM), Università di Messina, 98166 Messina, Italy; federico.nastasi@studenti.unime.it; 4Dipartimento di Agraria, Università degli Studi di Sassari, 07100 Sassari, Italy; pideiana@uniss.it (P.D.); msantona@uniss.it (M.S.); 5Laboratory of Chemistry, Department of Science, Agricultural University of Athens, 75 Iera Odos, 118 55 Athens, Greece; ptara@aua.gr; 6School of Chemistry & Physics, University of KwaZulu-Natal, Pietermaritzburg, Private Bag X01, Scottsville 3209, South Africa

**Keywords:** NMR analysis, RICC-NMR, extra virgin olive oil, *Olea europaea*, ^1^H{^13^C}-NMR, metabolic profile, PLS-DA

## Abstract

The paper describes the application of mono-dimensional high-resolution nuclear magnetic resonance (NMR) spectroscopy to extra virgin olive oil (EVOO) samples to extract the chemical composition reasonably featured by specific genotype (*cultivar*) and detailed environmental conditions (*terroir*). To achieve this goal, we designed a suitable spectroscopic protocol made up of four NMR experiments: (I) standard ^1^H{^13^C}, (II) multiple pre-saturated ^1^H{^13^C}, (III) ^1^H selective excitation at 9.25 ppm, and (IV) ^13^C{^1^H} acquisition. The three ^1^H-NMR experiments (I–III) were merged into a single informative ^1^H-NMR trace. This “Restored Intensities from Customized Crops” (RICC-NMR) allowed us to extract, in just one ^1^H-NMR dataset, combined information about (a) main components, (b) less-represented components, and (c) minor but key-ruling secoiridoid species, respectively. Selected integrations of the RICC-NMR trace, together with selected integrations gathered from the ^13^C-NMR profile, led, for each sample, to the quantification of variables able to sort out distinct EVOOs. In this paper, this original methodology was applied to rationalize eighteen EVOOs from Sicily and nineteen from Sardinia, which were chemometrically compared and discussed.

## 1. Introduction

Extra virgin olive oil (EVOO) is the non-polar fraction obtained by milling the drupes of *Olea europaea*. Its widespread consumption worldwide is chiefly due to the optimal balance among saturated (SFA), monounsaturated (MUFA), and polyunsaturated (PUFA) fatty acids and bioactive components [1]. While the high MUFA content significantly contributes to the health benefits of EVOO compared to other vegetable oils, its bioactivity is also influenced by a variety of minor compounds involved in key physiological pathways. These include aliphatic and triterpene alcohols, hydrocarbons, volatile compounds, squalene (SQ), sterols, phenols, and other bioactive molecules [2,3].

Among these minor components, lipophilic phenols (such as flavonoids, lignans, and secoiridoids) have gained significant attention due to their protective role against oxidative stress, demonstrated in both in vitro and in vivo studies [3,4,5,6]. Additionally, these phenolic compounds contribute to the self-stabilization of EVOO, enhancing its shelf life, and are also responsible for key organoleptic properties, particularly bitterness and aroma, despite their relatively low concentrations [7,8].

The characterization of EVOO relies on a range of analytical techniques, mainly employing separation methods coupled with highly sensitive detectors. Some of these methods are recognized as official protocols by the European Commission [9]. Gas chromatography with flame ionization detection (GC–FID) is the standard approach for quantifying fatty acids as glyceryl esters and other compounds [10,11]. High-performance liquid chromatography (HPLC), frequently coupled with diode array detection (DAD), is employed for the determination of specific triacylglycerols (triglycerides) [12] and alternatively for the quantification of phenolic compounds after extraction with acetonitrile [13]. Due to the complex nature of EVOO composition, several targeted analytical approaches have been developed to identify and quantify specific compounds [14].

Over the past decades, nuclear magnetic resonance (NMR) spectroscopy has gained prominence as a holistic approach [15]. The power of NMR lies in its pure quantitative response and direct sampling (without chemical treatments), which minimizes experimental errors and limits the potential use of standards [16]. The most stated limitation of this technique is sensitivity, linked to the relative concentration of chemicals with respect to the most represented species within the sample [17,18]. This is called “the dynamic range limit”, and specific targeted experiments can overcome this challenge through the attenuation or exclusion of the “giant” signals coming from the most represented compounds. In this study, we present an original experimental protocol to specifically tackle the mentioned dynamic range challenge, pushing the limits of NMR analysis of minor components of EVOOs. Specifically, beyond the application of ^13^C decoupling over a standard ^1^H-NMR experiment, we added a multi-suppressed variant enabling the free quantification of sterols and squalene, and a further selective experiment able to determine phenolic derivatives. These NMR results, simultaneously covering several chemical species, are not obtainable by any other traditional single device, and they are used to generate an approach able to interpret the differences between Sicilian and Sardinian EVOO samples, paving the way for other possible studies that embrace a wider plethora of samples.

## 2. Materials and Methods

### 2.1. Chemicals

High-purity deuterated chloroform (CDCl_3_, ≥99.8 atom % D, contains 0.5 wt. % silver foil as a stabilizer, catalog number 416754) and tetramethyl silane (TMS) were obtained from Merck (Milano, Italy).

### 2.2. Samples

As a case study to evaluate the application of our original procedure, as described in Section 2.3 and Section 2.4, we chose to compare two groups of extra virgin olive oils (EVOOs) from Sardinia (SAR) and Sicily (SIC), for which the literature provides genetic evidence indicating a certain degree of genetic correspondence [19]. The thirty-seven chosen EVOO samples were selected from Sicilian (eighteen samples of the *Nocellara* cultivar) and Sardinian (nineteen samples of the *Bosana* cultivar) crops to provide a rationale explaining the *Olea Europaea* terroir condition. Specific goods were provided by local producers in dark, sealed bottles after the 2023–2024 campaign. Samples were kept sealed in dark, dry cupboards (16 °C) until January 2025 and then picked up for analysis. Measurements were run in triplicate. After dilution in CDCl_3_, sample solutions were analyzed or kept in a −44 °C freezer. According to the NMR profile, the frozen solution remained stable for 20 days.

### 2.3. NMR Sample Preparation

Sample preparation was based on a specific strategy developed to seek the best compromise to obtain an acceptable field homogeneity for any sample, good sensitivity extended to the less-represented components and ^13^C signals, and a good separation between key signals of the ^13^C- NMR experiments (namely the 10-C of oleic and linoleic fragments) [20,21]. The most affected signals in ^13^C-NMR are the aromatic (unsaturated) *C*-H signals, whose shift precludes the serial alignment of spectra for feasible data processing [20]. The final samples were prepared by diluting original EVOO samples with CDCl_3_, keeping the relative weight ratio at 13.5:86.5; for quick and cost-effective preparation, this corresponds to adding 500 μL of deuterated chloroform to 128 μL of EVOO at 25 °C inside a 5 mm test tube for NMR analysis.

We remark again that other NMR strategies use more solvents and higher dilutions to achieve resolution; however, they lack specific sensitivity and insight into some of the mentioned degradable phenolic species [1]. In this context, increased sensitivity plays a crucial role in the detection of sterol species.

### 2.4. NMR Experimental Protocol

For all samples, four basic experiments were recorded:Experiment I: A standard ^1^H spectrum endowed with a ^13^C decoupling sequence during the acquisition, performed by 16 scans and a 17.2 s recycling delay for quantitative analysis.Experiment II: The same experiment I, with a multiple pre-saturated wave able to attenuate the main signals (NOESYGPPS). This experiment was run for 32 scans, with a 17.2 s recycling delay for quantitative analysis.Experiment III: The ^1^H- DPFGSE (selected double-pulsed field gradient spin echo) spectrum [21] was acquired with 40 scans for the determination of aldehydic–phenolic species, namely oleocanthal (TY-EDA, CAS Number: 289030-99-5), oleacein (HTY-EDA, CAS Number: 149183-75-5), ligstroside aglycone (HTY-EA, CAS Number: 174511-64-9), oleuropein aglycone (TY-EA, CAS Number: 31773-95-2), and elenolide (ELNL, CAS Number: 24582-91-0).Experiment IV: Full-time ^1^H-decoupled ^13^C spectrum with 128 scans, with a suitable recycling delay for quantitative evaluations (more than 18 s per cycle) [20].

This experimental protocol resulted from our personal optimization, aimed at achieving the best performance within a reasonable overall experiment time. The four experiments’ setups lasted around 5 min, 10 min, 9 min, and 40 min, respectively; therefore, the total experimental analysis for every sample was 90 min in the worst-case scenario.

### 2.5. NMR Acquisition and Processing

Samples in the 5 mm NMR tubes were analyzed by a 600 MHz Bruker (Bruker Biospin, Milan, Italy) spectrometer equipped with a BBI probe with gradients at the constant temperature of 25 °C. After the automatic tuning and gradient shimming, the line shape of the CHCl_3_ residual signal was checked by shimming until the line shape was lower than 1.0 Hz. NMR spectra for the ^1^H and ^13^C nuclei were run at 600.13 and 150.73 MHz, respectively. The hard pulse for the maximum sensitivity (90° pulse) was calibrated throughout the samples and was always within 8.2 ± 0.1 μs at −11.3 dB. Selective, multi-suppression (NOESYGPPS) and ^13^C{^1^H} experiments were conducted as previously described [22]. The ^1^H spectra (I, II, III) were acquired with the same spectral width (12 ppm), acquisition time (AQ = 2.2 s), and delay (15 s).

Experiments I, II, and III were run with a spectral width of 12 ppm, 64 scans, 2 s of acquisition time, and 15 s of recycle delay to ensure that the quantitative methods remained valid regardless of the different proton relaxation times (maximum value of *T*_1_ = 2.5 s, which is less than one-fifth of the total recycling time), resulting in a total acquisition time of 24 min.

By following our previous studies, we set up experiment III (^1^H-DPFGSE pulse sequence) [23]. The shaped pulse, with an offset centered at a frequency of 9 ppm, set to excite the signals in the 8–10 ppm range, was an optimized BURP (Band-Selective, Uniform Response, Pure-Phase) [24] with a duration of 7 ms. Since in the first 8 or 16 scans, the signal-to-noise ratio was good enough for spectral lines to begin to be observed, Experiment III is promising for better detection and quantification of the aldehydic secoiridoid derivatives. The final spectra after 40 scans were processed after a slight line-broadening exponential window function (0.3 Hz) through the Fourier transform procedure.

For Experiment II (NOESYGPPS), a ^1^H NMR pulse sequence was employed. Consistent with a previous study [22], an amplitude- and phase-modulated shaped pulse was applied during the acquisition time, comprising 29 highly selective frequency bands, each with a bandwidth of 5 Hz. This enabled the highly selective suppression of dominant lipid signals while preserving the remainder of the spectrum undistorted within ±0.1 ppm of each suppressed signal.

For the same reason, Experiment IV (^13^C{^1^H}) was first acquired with the ^13^C 90° hard pulse (15 ± 0.3 μs at −20.2 dB, 112 W), 128 scans, 2 s of acquisition time, and 20 s for the relaxation delay. Afterward, these experiments were compared to others with lower recycling delay and smaller tilting angles with the purpose of optimizing the experimental time, while keeping the quantitative ratio of the ^13^C signals. The final optimized conditions were as follows: 128 scans, 82° pulse, and 20 s of recycling delay for a total of 64 min of experimental time.

The calibration of frequencies in Experiments I, II, and III was basically performed on the methyl group of the β-sitosterol signal at δ_H_ = 0.738 ppm (previously set with respect to the TMS δ_H_ = 0.0 ± 0.005 ppm). Similarly, for ^13^C frequency calibration (Experiment IV), the glycerol CH signal (>C*H*-OH; δ^13^C = 78.9340 ppm, previously set for the TMS δ_c_ = 0.0 ± 0.005 ppm) was used. This provided a TMS-free frequency calibration of oil samples, eliminating many experimental issues.

### 2.6. NMR Processing Strategies and Quantification

Our idea to collect different ^1^H experiments (I, II, and III) aimed to achieve suitable precision in peak integration in a completely different dynamic range, reflecting a wide relative concentration of the EVOO components. To handle a limited dataset, we had the idea to rebuild just one ^1^H NMR profile cropped out from the three experiments, whose retained regions retain the optimized sensitivity of that specific experiment. The original “Restored Intensities from Customized Crops of NMR experiments” (RICC-NMR) saves, in just one spectral trace (Figure 1), the best sensitivity for the main components (taken from Experiment I), less-represented components (sourced from Experiment II, which is also the quantitative reference for the other insets), and minor phenolic fraction in the 8.5–10 ppm range. The serial processing procedure, including alignment by Icoshift [25], targeted integrations of the RICC-NMR profile, and the ^13^C NMR experiments, were performed using MATLAB (Matlab R2024a software package—The Mathworks, Cambridge, UK). Specific command lines and procedures are explained in the Appendix A.

The integration of 102 regions from the ^1^H RICC-NMR profile and 98 regions from ^13^C-NMR enabled the optimized quantification of 19 variables (reported in Table 1) according to the assessed MARA-NMR algorithm [26]. The mentioned integrations target specific and significant signals whose intensity is due to one or more chemical groups belonging to parent chemical species. Tables reporting the specific integration ranges, assignments along with spectral traces, and graphical assignments are reported in Appendix A. This information was first used to assess the quality of the EVOOs and rationalize the statistical trends. Specifically, the evaluation of peroxides and di-acyl glycerols, along with the general observation of the other “in-range” compounds, were screened. Briefly, the MARA-NMR optimization procedure [26] is based on accounting for the overall spectral contribution sorted out from any chemical in a mixture. It produced 21 equations from the whole ^1^H NMR profile and 73 equations from ^13^C NMR, resulting in the quantification of 19 significant components. The precision of this quantification was assessed through measurements performed in triplicate across three separate days, yielding nine values for each variable. MARA-NMR is based on the least-squares regression applied to the global data; therefore, low quadratic deviation values were considered a guarantee of self-consistent outcomes.

The unmatched holistic NMR analytical power spans the simultaneous determination of different classes of compounds, such as fatty glyceryl esters, sterols, and phenolic derivatives (Table 1 and Appendix A).

The choice of just nineteen variables stems from the evaluation of the single-variable precision, statistical significance, and exclusion of parameters affected by EVOO shelf life (peroxides and di-acyl glyceryl esters), as these may be affected by non-standardized or undefined storage conditions.

### 2.7. Statistical Analysis

Statistical analyses were performed using the SIMCA-P software package version 14.1 (Umetrics AB, Umea, Sweden), and data were scaled using the univariate procedure. Principal component analysis (PCA) data analysis was performed for exploratory purposes and outlier detection, while projection to latent structures (PLS)-based methods were employed for discriminant analysis and dataset comparison. We used an orthogonal extension of PLS-DA [27], where the first latent variable only accounted for variations in correlated data. PLS-DA models were evaluated using the goodness-of-fit parameter (R2Y) and the predictive ability parameter (Q2Y). First, 37 independent samples were chosen to perform a multivariate statistical analysis without a defined classification. Due to the relatively limited available samples, we opted to use the nineteen variables output by NMR quantification to avoid overfitting, collinearity, and challenging recovery of the chemical rationale. The results were analyzed, and their robustness was verified using the methods proposed by Erikson [28]: a permutation test was performed by randomly changing the class assignment of 3 samples for time.

## 3. Results

### 3.1. Data Processing and Treatment

Processing of the mentioned NMR data resulted in the quantification of 19 components, which are listed in Table 1. In the Appendix A, the molecular structure of the quantified compounds is reported.

### 3.2. Statistical Analysis of the Metabolic Profile

We selected a dataset that would allow us to test the feasibility of using RICC-NMR as a rapid method to obtain analytical data that would otherwise require multiple extraction and quantification techniques. On the other hand, the *Olea europaea* varieties present in Sicily and Sardinia share an assessed genotype origin [19], likely due to the historical patterns of olive diffusion across the Mediterranean Sea. Starting from this hypothesis, we highlighted the differences presumably attributable to pedoclimatic effects.

Statistical graphs are presented in Figure 2. For the statistical analysis, we used 37 observables and 19 variables. The PCA in Figure 2a shows a promising clustering of samples according to specific variables, revealing a fair distribution that may be linked to several factors (i.e., temperature, altitude, solar exposure, and/or pedologic factors). The cumulative sum of squares R^2^ (0.554) and the fraction of the total variation of X, Q^2^ (0.139) indicate insufficient fit, which may be due to the low number of samples. The PCA shows clustering into two groups: SIC (blue) and SAR (red) (Figure 2a).

The partial least squares discriminant analysis (PLS-DA) model (R^2^ = 0.889; Q^2^ = 0.789) demonstrated a robust and reliable separation between the SIC and SAR sample groups (Figure 2c). The model’s robustness was evaluated through both leave-one-out cross-validation and random permutation testing.

In PLS-DA, permutation testing assesses the statistical significance of the model by randomly shuffling class labels and determining whether the model’s performance metrics remain robust, thus mitigating the risk of overfitting. It displays the correlation coefficient between the original y-variable and the permuted y-variable on the *x*-axis versus the cumulative R^2^ and Q^2^ on the *y*-axis and draws the regression line. The intercept is a measure of the overfit. A 300-fold permutation test was conducted, resulting in regression lines with R^2^ and Q^2^ intercepts of 0.247 and −0.305, respectively, supporting the model’s validity. These results confirm that the PLS-DA model is both statistically robust and suitable for extension to larger or future datasets.

We also used an orthogonal extension of OPLS-DA [27], in which the first latent variable accounted only for correlated data variations, filtering out the variation not directly related to the discriminant response (Figure 2b). A complete separation of the two classes was obtained.

In Figure 2b, we can identify the most important variables and their roles in the PLS-DA model. These are PHGR, VSTR2, O2, VSTR, ELNL, Ln, O+PO, and HTY-EDA. The results are consistent with those obtained by Piravi-Vanak [29], who demonstrated that the fatty acid and sterol composition of EVOO is strongly influenced by the variety, ripening process, and geographical origin, particularly the latitude and climatic conditions [30].

In Figure 2c, the variable importance in projection (VIP) score is shown as a weighted sum of the squared PLS weights (w*), representing the cumulative measure of the influence of individual X-variables on a PLS model. Usually, VIP scores larger than 1 are the most relevant for explaining the dependent variable vector, Y.

The classification performance was further evaluated through a misclassification test (Table 2), which reports the number of observations correctly classified into their respective groups using leave-one-out cross-validation. As Table 2 shows, all the samples were accurately classified.

## 4. Discussion

Chemometrics is the science of extracting information from chemical systems by data-driven means, aiming to interpret variations and differences across various categories using specialized mathematical and statistical tools. Although this can be accomplished by acquiring all spectral information (through bins), we believe that the selection of a specific target enables a simpler discussion of target effects free from artifacts. The quantification of minor components is the most effective approach for understanding the mechanisms of differentiation driven by genotypic and/or pedoclimatic conditions [31]. Most published papers adopt the continuous bucketing approach (e.g., 0.04 ppm), subsequently attempting to identify a component responsible for the specific discrimination (a fully untargeted analysis). Nevertheless, this approach is affected by dataset uncertainties related to data non-collinearity across different buckets [32]. The use of different spectra enabled the extraction of targeted, extensive information on minor components through definite integrals, which were typically obscured by more intense signals within standard NMR experiments. Our multi-experimental approach takes advantage of the best features provided by the different techniques; for instance, the multi-suppression of main signals and simultaneous ^13^C decoupling cleans several regions where tiny but crucial signals are fairly displayed.

Such selected datasets enabled a more confident application of statistical analyses, providing immediate insights into the influence of climatic conditions.

For this reason, we selected two cultivars belonging to the same *Phoenician* botanical family [19], cultivated on two major islands located in the central Mediterranean region.

The 19 chosen variables (Table 1) were identified among 31 quantification indicators. Five variables were strictly related to the EVOOs’ shelf-life (namely di-acyl glyceryl esters and peroxides), but this topic falls outside of the purpose of the specific work. However, these variables were clearly randomly distributed and therefore not used in the discriminant analysis. According to their statistical significance, seven other variables were excluded.

Through this vision, the composition of polyphenols and sterols directly accounts for the response of olive trees to climatic conditions. Specifically, during the 2023–2024 season, despite higher temperatures in northern Sardinia compared to the average of previous years [33], the comparison with climatic conditions in Sicily clearly demonstrates that in Sardinia, lower temperatures and higher humidity led to higher production of some polyphenols (oleuropein and ligstroside aglycone) and sterols [34].

OPLS-DA identified main differentiators based on geographical region; vegetal sterols and ligstroside aglycone characterize the Sardinian oils, whereas Sicilian samples are correlated positively with oleate esters and linoleate esters. These main discriminant components confirm the previously observed climatic distinction evidenced in earlier studies [20,30].

As reported by Lukic et al. [35], the influence of agronomic, varietal, and pedoclimatic variables is complex, affecting the composition of fatty acids, sterols, and other minor components. In this paper, we highlight the dominant influence of climatic conditions on the composition of polyphenols and sterols in EVOOs. Specifically, higher temperatures are associated with an increase in the production of both free and glycerol-bound fatty acids, while the composition of sterols and polyphenols is negatively affected. This effect can be interpreted as a plant response mechanism to environmental stress induced by climatic conditions [30].

## 5. Conclusions

This study consisted of the development of a customized NMR analytical protocol for EVOO samples, including (a) a tailored sample preparation, (b) an experimental workflow comprising three distinct ^1^H-NMR acquisitions and one ^13^C-NMR acquisition, (c) an original data processing approach (using Mestrenova v15 release 2023, Santiago de Compostela, ES, Europe; and MATLAB R2024a software package—The Mathworks, Cambridge, UK) based on the alignment and integration of the three ^1^H-NMR experiments (RICC-NMR) along with the ^13^C-NMR profile, and (d) a targeted integration analysis to determine the compositional profile of olive oil samples (MARA-NMR). This original throughput was applied to Sicilian and Sardinian samples known to belong to the same *Olea europaea* strain [19,36,37]. Statistical chemometric methods allowed us to distinguish sample classifications and discuss the differences, which were mostly related to geographical origin.

This NMR approach targeting minor components demonstrates the key role of such species in characterizing the origin and features of complex food matrices such as olive oil, and it may pave the way for wider chemometric analyses. Moreover, this protocol is extendable to innovative studies on biological and environmental samples.

## Figures and Tables

**Figure 1 foods-14-01807-f001:**
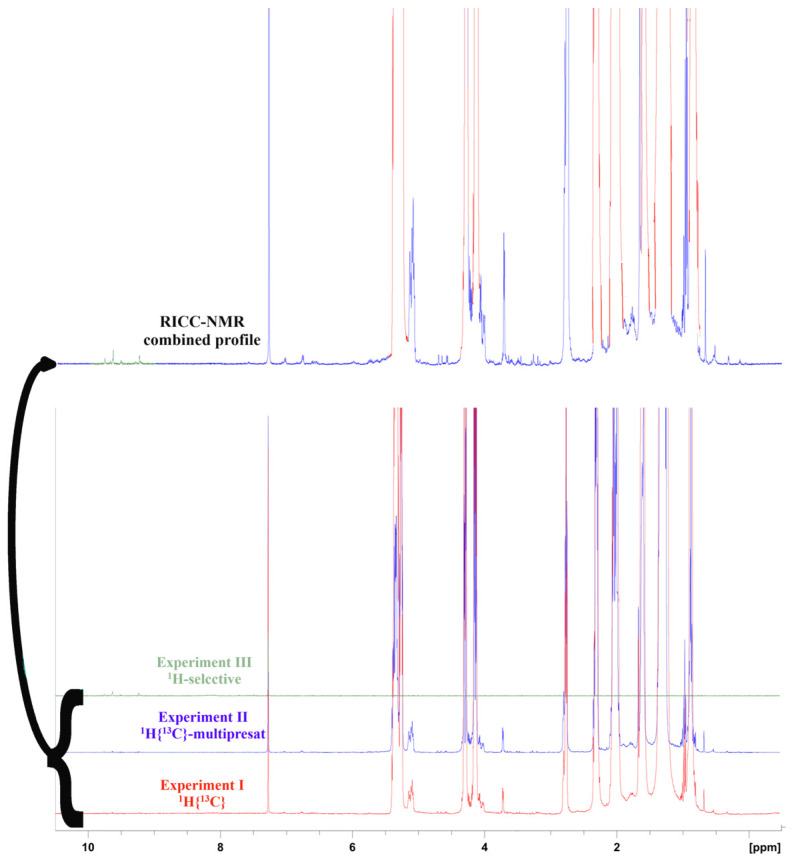
The ^1^H{^13^C} (Experiment I), ^1^H{^13^C} multi-presaturated over main signals (Experiment II), and the 8.5–10.5 ppm band-selective 1H experiments (Experiment III) are combined into a single trace, called RICC-NMR, retaining the most valuable information from each interesting region.

**Figure 2 foods-14-01807-f002:**
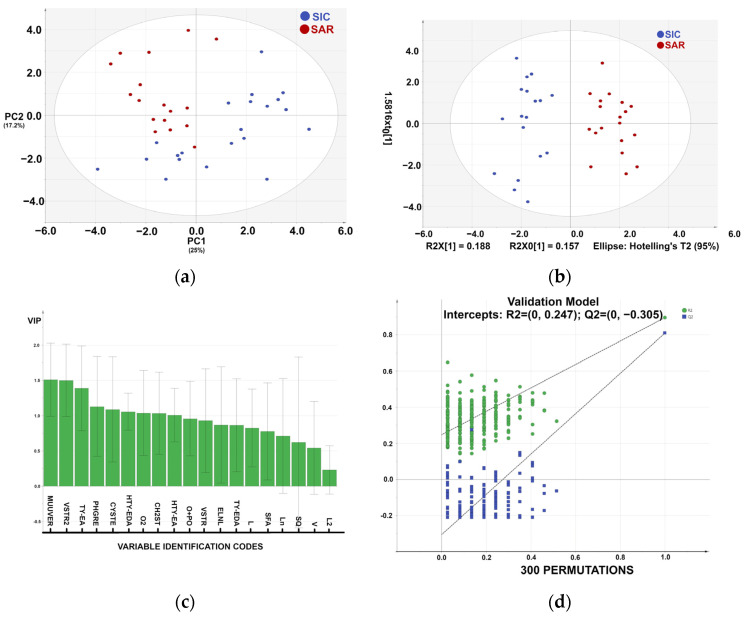
Statistical graphs. (**a**) PCA (principal component analysis) plot was constructed for exploratory purposes and outlier recognition. The plot shows a promising clustering trend. (**b**) The OPLS-DA plot model, which successfully discriminates samples of different origins. (**c**) VIP (Variable importance in projection) plot against the variables on the horizontal axes. (**d**) Permutation plot; the cluster of points on the left represents 300 permuted R^2^ and Q^2^ values. Green and blue dots correspond to R^2^ and Q^2^, respectively. The dashed lines indicate the fitted regression lines for both the observed and permuted R^2^ and Q^2^ values.

**Table 1 foods-14-01807-t001:** The chemical names and quantification units of the nineteen components used, with their abbreviation codes, measured mean values, and relative standard deviations.

	Quantified Compound(s) (Units *)	Code	Average Values	SD
1	Squalene (mol ppm)	SQ	1514.994	398.483
2	Linolenate esters (%)	Ln	0.745	0.098
3	Linoleate esters (%)	L	9.856	1.245
4	Oleate and Palmitoleate esters (%)	O+PO	66.291	3.089
*5*	*cis*-Vaccenate esters (%)	V	3.765	0.546
6	Palmitate and Stearate esters (%)	SFA	19.343	2.344
7	2-Glyceryl linoleate esters (%)	L2	4.743	0.588
8	2-Glyceryl oleate esters (%)	O2	26.111	1.190
9	β-Sitosterol, Δ^5^-avenasterol, Δ^5^-Campesterol (mol ppm)	VSTR	2503.748	335.051
10	Cycloeucalenol, 24-methylene cycloartanol, gramisterol (mol ppm)	CH2ST	842.772	390.310
11	Esters of cycloartenol, 24-methylene cycloartanol, and cyclobranol (mol ppm)	CYSTE	618.625	235.650
12	Citrostadienol, Δ^7^-avenasterol, Δ^7^-campesterol (mol ppm)	VSTR2	479.817	115.195
13	Maslinic and ursolic acid, uvaol, and erythrodiol (mol ppm)	MUUVER	1664.113	415.099
14	Phytol and geranylgeraniol esters (mol ppm)	PHGRE	410.641	159.199
15	Oleocanthal (mol ppm)	TY-EDA	397.855	201.281
16	Oleaceine (mol ppm)	HTY-EDA	252.839	166.735
17	Ligustroside aglycone (all the derivatives) (mol ppm)	TY-EA	163.956	150.862
18	Oleuropein aglycone (all the derivatives) (mol ppm)	HTY-EA	116.228	105.437
19	Elenolide (mol ppm)	ELNL	54.802	170.958

* As measured by NMR experiments, quantification units are in molecular ratios. This is why we used the molecular % for major components and mol ppm as the number of specific molecules over one million random molecules in the EVOO.

**Table 2 foods-14-01807-t002:** Misclassification test summarizing the number of observations with known class memberships that were correctly classified in the class or PLS-DA models. All samples are correctly classified.

	Members	Correct	SIC	SAR
SIC	18	100%	18	0
SAR	19	100%	0	19
Total	37	100%	18	19
Fisher’s prob.	5.7 × 10^−11^			

## Data Availability

The original contributions presented in this study are included in the article/Appendix A. Further inquiries can be directed to the corresponding authors.

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
