# Peer review of "Restored Intensities from Customized Crops of NMR Experiments (RICC-NMR) to Gain Better Insight on Chemometrics of Sicilian and Sardinian Extra Virgin Olive Oils"

_foods, 2025, doi:10.3390/foods14101807_

Round 1

Reviewer 1 Report

Comments and Suggestions for Authors

In this manuscript, four different NMR experiments were used to evaluate the Sicilian and Sardinian Extra Virgin Olive Oils components. It is interesting research with meaningful results, however, needs some revision before acceptance. 

Delete (RICC-NMR) from the title and also point mark from end of the title

Line 27, ... very tiny... replace it with minor

Line 43, fatty acids [1].  should be fatty acids and bioactive components [1]. 

Line 48, please give examples for lipophilic phenols. As authors know hydrophilic phenols is more pronounced in virgin olive oils than lipophilic phenols.

Line 90-92, Chemical section, please replace it and should be in section 2.1 and samples should be next.

Line 167-173, and 206-213 should be in discussion part. It is not related to the methods section.

It would be better to show Quantified Compound in spectra as a figure. 

Please discuss it if this method can detect adulteration with VOO from other regions or other vegetable oils as well or not. If yes, also, it would be interesting to give lower quantity which could be detected.

Author Response

We are very thankful to the reviewer for his/her suggestions and indications which hopefully would improve the quality of the paper. We list below our best response (in red) to all the formulated comments.

Regards

Comment1:

In this manuscript, four different NMR experiments were used to evaluate the Sicilian and Sardinian Extra Virgin Olive Oils components. It is interesting research with meaningful results, however, needs some revision before acceptance. 

Delete (RICC-NMR) from the title and also point mark from end of the title

Response 1:

We are very thankful to the reviewer for the constructive observations. According to the suggestion of another reviewer (number 4) we have kept the RICC acronym and rewritten the abstract.

Comment 2:

Line 27, ... very tiny... replace it with minor

Line 43, fatty acids [1].  should be fatty acids and bioactive components [1]. 
Response 2:

Thankfully we have edited the main text.

Comment 3:

Line 48, please give examples for lipophilic phenols. As authors know hydrophilic phenols is more pronounced in virgin olive oils than lipophilic phenols.

Line 90-92, Chemical section, please replace it and should be in section 2.1 and samples should be next.

Response 3:

We have changed text doing our best to improve the paper quality.

Comment 4:

Line 167-173, and 206-213 should be in discussion part. It is not related to the methods section.

It would be better to show Quantified Compound in spectra as a figure. 

Response 4:

We have done our best by following these indications merged with other observations (especially from reviewer n° 2). The whole discussion concerning the quantified compounds was rewritten, mentioning new figures added to the supplementary material.

Comment 5:

Please discuss it if this method can detect adulteration with VOO from other regions or other vegetable oils as well or not. If yes, also, it would be interesting to give lower quantity which could be detected.

Response 5:

Thank you. Paragraph 2.6 was totally rewritten stating that 1H- NMR profile is providing general holistic considerations on the olive oil samples and some targeted signals are providing the quantification of peroxides and di acyl glycerides which might give  some idea of the product quality, however this is not the aim of the paper and we could not find the room to give a better insight on this topic so it is not further detailed. We also have stated it in the new paragraph added to the discussion section.

Reviewer 2 Report

Comments and Suggestions for Authors

This study arouses interest in the title, the abstract and even in the methodology. However, when it came to the results, I was overcome with disappointment.

Lot of work was performed as explained in material and methods section, however none of this is commented neither discussed. Moreover, results are poor, and scientific errors can be observed.

Important comments:

  • How were combined the spectra resulted from the four experiments is not explained. Was it by summing all? Outer product? – please specify in the section 2.6
  • How is “MARA-NMR” applied? On the combined 4 experiments? One by one experiment? Also, how this method works? This is not explained in the section 2.6
  • Traditional analyses were performed by official methods, but results are not presented, neither commented, neither discussed. If they were not used, please remove. If they were used, please present and comment. Note that free fatty acidity (FFA) is not the correct term according to EU legislation.
  • On which data were PCA and PLS-DA methods applied? This must be explained in section 2.8, even if later is commented in results. In results it is explained that 19 variables were the input data, but it was related to the intensity of each peak in the NMR spectra? Or the calculated concentration appearing in table 1?
  • R2 of 0.247 can never “support the model”, is not a good fit. Also, what is Q2 = -0.305 interpretability? What implies the negative number on this parameter? What should it be? Please explain in section 3.2. Moreover, “suitable for extension to larger or future datasets” (line 256) can never be confirmed if no external validation with new samples was performed.
  • Line 261. “From figure 2b, we can identify the most important variables” is not true, maybe it is for figure 2c.
  • Figure 2: (a) should represent scores from one PC versus other PC, please specify in the axes with the cumulative variance % explained by each PC. (b) what are axes representing? Difficult to understand – is this figure plotting the PLS-DA or the OPLS-DA results? (contradictory info on lines 246 and 259). (c) and (d) what are axes? What are colours on 2(d)? please provide a good-size legend.
  • Lines 273-276. What do you mean with “each dimension” ? what is Y? this should be identified for the reader
  • Discussion section: “Nevertheless, this approach is affected by dataset uncertainties related to data non-collinearity across different buckets.” Can you provide references supporting this?
  • Line 297. As far as I understood, the PLS-DA model was developed to discriminate samples with different geographical origin, nothing about climatic conditions was commented on results. Were oils coming from the same olive variety? Maybe this is the reason for the different minor components composition – was this considered?
  • “… higher humidity lead to a higher production of some polyphenols (Oleuropein and Ligstroside aglycone) and sterols” – this is not supported in the study.
  • Why the 19 components (table 1) were the selected? How were they found in the data? – this information is also missing in the manuscript. Moreover, why these components were not measured following the official methods recognised from EU regulation to compare the results instead of analysing other parameters that finally were not used in the study?

In short, I see a great lack of information and many gaps that make it difficult to understand either the objective of the study or how it was carried out, or what it contributes to the world. Because, is it really necessary to spend 90 minutes of analysis in an NMR that has a very high cost and maintenance requirements to then put all the data together, to be left with only 19 peaks and all this just to find out if the oil comes from Sardinia or Sicily? For me, the article should be much improved but I am willing to give the authors a second chance.

Other minor comments:

  • Contradictory four (line 22) and then three (line 25) experiments. “sample chemometrics” this term makes no sense, lilne 33.
  • Line 86: and storage temperature? EVOO is susceptible of oxidation
  • Please use ºC instead of K for temperature, or at least homogenize because both are used in the manuscript
  • Please make sure what was the measurement time: 8, 8, 8, 40 min (line 124), 18 min (line 141) and 48 min (line 159)
  • Line 132: specify these frequencies are for 1H and 13C respectively
  • Line 136: 12 ppm spectral width for 13C spectrum ???
  • Line 161: what do you mean with calibration?
  • Line 167. This is not a correct definition of chemometrics
  • Line 219: PLS is partial least squares / define DA in PLS-DA (line 221)
  • Table 1: mol ppm units is totally incorrect. If its mol, is not concentration but number, if it is ppm, is concentration. But in any case, no sense makes both together. Also, if most of the components are in %, all of them should be equal. / average value? Of what? All the samples? Why was not present by groups, e.g. according to the geographic origin and comment the differences, should be interesting.
  • Line 236: metabolic profile doesn’t seem a correct term in this context, as it is used for human blood analysis.
  • Line 243. Define SIC and SAR
  • Line 298. Term Phoenician doesn’t look correct in this context.

Author Response

We are very thankful to the reviewer for his/her suggestions and indications which hopefully would improve the quality of the paper. We list below our best response (in red) to all the formulated comments.

Regards

Comments 1:

This study arouses interest in the title, the abstract and even in the methodology. However, when it came to the results, I was overcome with disappointment.

Lot of work was performed as explained in material and methods section, however none of this is commented neither discussed. Moreover, results are poor, and scientific errors can be observed.

Important comments:

  • How were combined the spectra resulted from the four experiments is not explained. Was it by summing all? Outer product? – please specify in the section 2.6
  • How is “MARA-NMR” applied? On the combined 4 experiments? One by one experiment? Also, how this method works? This is not explained in the section 2.6

Response 1:

 We thank the reviewer for the important considerations, and the entire paragraph 2.6 was rewritten following your indications

Comments 2:

  • Traditional analyses were performed by official methods, but results are not presented, neither commented, neither discussed. If they were not used, please remove. If they were used, please present and comment. Note that free fatty acidity (FFA) is not the correct term according to EU legislation.

Response 2:

Thanks for the suggestions. Paragraph was deleted.

Comments 3:

  • On which data were PCA and PLS-DA methods applied? This must be explained in section 2.8, even if later is commented in results. In results it is explained that 19 variables were the input data, but it was related to the intensity of each peak in the NMR spectra? Or the calculated concentration appearing in table 1?
  • R2 of 0.247 can never “support the model”, is not a good fit. Also, what is Q2 = -0.305 interpretability? What implies the negative number on this parameter? What should it be? Please explain in section 3.2. Moreover, “suitable for extension to larger or future datasets” (line 256) can never be confirmed if no external validation with new samples was performed.

Response 3:

Dashed lines represent the fitted regression lines corresponding to the observed and permuted R² (green dots) and Q² values (blue dots).

 As reported in the SIMCA-P help, “The purpose of this validation is to compare the goodness of fit (R² and Q²) of the original model with that of several models generated by randomly permuting the order of the Y-observations, while keeping the X-matrix unchanged.

Comments 4:

  • Line 261. “From figure 2b, we can identify the most important variables” is not true, maybe it is for figure 2c.

Response 4:

The typing error was corrected

Comments 5:

  • Figure 2: (a) should represent scores from one PC versus other PC, please specify in the axes with the cumulative variance % explained by each PC. (b) what are axes representing? Difficult to understand – is this figure plotting the PLS-DA or the OPLS-DA results? (contradictory info on lines 246 and 259). (c) and (d) what are axes? What are colours on 2(d)? please provide a good-size legend.

Response 5

Thank you, we have modified Fig2a, caption and main text accordingly

Comment 6:

  • Lines 273-276. What do you mean with “each dimension”
  • ? what is Y? this should be identified for the reader

Response 6:

The sentence was changed to explain this concept

Comment 7:

  • Discussion section: “Nevertheless, this approach is affected by dataset uncertainties related to data non-collinearity across different buckets.” Can you provide references supporting this?

Response 7:

We have added the reference [32] changing all the other calls accordingly

Comment 8:

  • Line 297. As far as I understood, the PLS-DA model was developed to discriminate samples with different geographical origin, nothing about climatic conditions was commented on results. Were oils coming from the same olive variety? Maybe this is the reason for the different minor components composition – was this considered?

Response 8:

Thanks to your suggestions we have added a paragraph in the 3.2 section and the overall discussion (section 4) was modified accordingly

Comments 9:

  • “… higher humidity lead to a higher production of some polyphenols (Oleuropein and Ligstroside aglycone) and sterols” – this is not supported in the study.

Response 9:

The reference https://doi.org/10.1016/j.eja.2025.127506 was added

Comments 10:

  • Why the 19 components (table 1) were the selected? How were they found in the data? – this information is also missing in the manuscript.

Response 10:

According to the suggestions of you and other reviewers we have clarified this aspect in the last paragraph of 2.6 and ion the main discussion

The variables were selected based on the ratio of standard deviation to mean (std. dev./mean < 0.3)."

Comment 11:

  •  Moreover, why these components were not measured following the official methods recognised from EU regulation to compare the results instead of analysing other parameters that finally were not used in the study?

Response 11:

Thankfully we have erased the section 2.7

Comments 12:

In short, I see a great lack of information and many gaps that make it difficult to understand either the objective of the study or how it was carried out, or what it contributes to the world. Because, is it really necessary to spend 90 minutes of analysis in an NMR that has a very high cost and maintenance requirements to then put all the data together, to be left with only 19 peaks and all this just to find out if the oil comes from Sardinia or Sicily? For me, the article should be much improved but I am willing to give the authors a second chance.

Response 12:

By thanking the reviewer, we tried our best to fulfil the evidenced gaps which hopefully would improve the quality of the manuscript. Specifically, it is explained how 19 chosen variables obtained through this original NMR multi-experimental protocol without massive solvent waste are not simultaneously detectable by any other analytical technique in a single run. Moreover, as demonstrated by the ref 21 (in the manuscript) CDCl3 NMR is fairly detecting phenols which are changing their chemical nature in polar media precluding the straightforward use of RP-HPLC. We point out that this approach turned out to be successful on the presented case-study

Comments 13:

Other minor comments:

  • Contradictory four (line 22) and then three (line 25) experiments. “sample chemometrics” this term makes no sense, lilne 33.
  • Line 86: and storage temperature? EVOO is susceptible of oxidation
  • Please use ºC instead of K for temperature, or at least homogenize because both are used in the manuscript
  • Please make sure what was the measurement time: 8, 8, 8, 40 min (line 124), 18 min (line 141) and 48 min (line 159)
  • Line 132: specify these frequencies are for 1H and 13C respectively
  • Line 136: 12 ppm spectral width for 13C spectrum ???
  • Line 161: what do you mean with calibration?
  • Line 167. This is not a correct definition of chemometrics
  • Line 219: PLS is partial least squares / define DA in PLS-DA (line 221)
  • Table 1: mol ppm units is totally incorrect. If its mol, is not concentration but number, if it is ppm, is concentration. But in any case, no sense makes both together. Also, if most of the components are in %, all of them should be equal. / average value? Of what? All the samples? Why was not present by groups, e.g. according to the geographic origin and comment the differences, should be interesting.
  • Line 236: metabolic profile doesn’t seem a correct term in this context, as it is used for human blood analysis.
  • Line 243. Define SIC and SAR
  • Line 298. Term Phoenician doesn’t look correct in this context.

Response 13:

All the observations were considered and fixed to our best by changing sentences and paragraphs.

Reviewer 3 Report

Comments and Suggestions for Authors

Dear authors,

I have evaluated your manuscript entitled “Restored Intensities from Customized Crops of NMR experiments (RICC-NMR) to gain a better insight on chemometrics of Sicilian and Sardinian Extra Virgin Olive Oils.” The topic is highly relevant, and the proposed multi-experiment RICC-NMR approach could enrich the analytical toolbox for EVOO authentication. Nevertheless, a number of serious concerns must be satisfactorily addressed before the work can proceed further in the editorial process.

Correct the discrepancy in sample numbers. The abstract states that four NMR experiments were run “on the same olive-oil sample”, whereas §2.1 describes 37 distinct EVOOs. Revise the abstract to reflect the true scope of the study or clarify if the single-sample statement is intentional (e.g., for methodological development only).

Section 2.2 (Chemicals) – Expand the section: supplier, catalogue number, stated purity, isotope enrichment (for CDCl₃), residual water and stabiliser content.
Section 2.4 (NMR protocol) – Replace qualitative phrases such as “suitable cycling delay” with explicit values for each experiment (recycle delay, power level, pulse sequence name/version, acquisition time, dwell time, receiver gain).
Clarify ambiguous terminology: what exact structural class is implied by “aldehydic-phenolic species”? Provide IUPAC names or CAS numbers for at least the major analytes.

Provide enough representative spectra. Include in the main text or Supplementary Material: (i) full 1H traces for experiments, (ii) the 13C spectrum, and (iii) annotated regions highlighting the quantified signals with peak assignments, integration limits, etc.

Supply the MATLAB scripts used for shift alignment, NMR quantification and statistical modelling, or deposit them in a public repository (GitHub/Zenodo) with a DOI.
Justify the choice of integrating 19 variables.

Ensure that the file referenced are uploaded and contains at minimum: raw FIDs, processed spectra, code, integrals, climate data for the campaign, etc.

A proofreading pass is needed, as form example “secoridoid” should be “secoiridoid” “worthed information”, replace non-standard abbreviation “MUF9” with the accepted lipid shorthand, some journal abbreviations and italics inconsistencies, etc.

Given the scientific interest of the manuscript and the fact that all shortcomings are, in principle, curable by rewriting, re-analysis and improved documentation, I recommend major revision. Please address each comment point-by-point and provide a tracked-changes version of the manuscript.

I look forward to evaluating a thoroughly revised submission.

Sincerely,

Comments on the Quality of English Language

Must be improved.

Author Response

We are very thankful to the reviewer for his/her suggestions and indications which hopefully would improve the quality of the paper. We list below our best response (in red) to all the formulated comments.

Regards

Comments 1:

I have evaluated your manuscript entitled “Restored Intensities from Customized Crops of NMR experiments (RICC-NMR) to gain a better insight on chemometrics of Sicilian and Sardinian Extra Virgin Olive Oils.” The topic is highly relevant, and the proposed multi-experiment RICC-NMR approach could enrich the analytical toolbox for EVOO authentication. Nevertheless, a number of serious concerns must be satisfactorily addressed before the work can proceed further in the editorial process.

Correct the discrepancy in sample numbers. The abstract states that four NMR experiments were run “on the same olive-oil sample”, whereas §2.1 describes 37 distinct EVOOs. Revise the abstract to reflect the true scope of the study or clarify if the single-sample statement is intentional (e.g., for methodological development only).

Response 1:

The whole abstract was rewritten in order to clarify several points. We are very grateful to the reviewer

Comments 2:

Section 2.2 (Chemicals) – Expand the section: supplier, catalogue number, stated purity, isotope enrichment (for CDCl₃), residual water and stabiliser content.

Section 2.4 (NMR protocol) – Replace qualitative phrases such as “suitable cycling delay” with explicit values for each experiment (recycle delay, power level, pulse sequence name/version, acquisition time, dwell time, receiver gain).
Clarify ambiguous terminology: what exact structural class is implied by “aldehydic-phenolic species”? Provide IUPAC names or CAS numbers for at least the major analytes.

Provide enough representative spectra. Include in the main text or Supplementary Material: (i) full 1H traces for experiments, (ii) the 13C spectrum, and (iii) annotated regions highlighting the quantified signals with peak assignments, integration limits, etc.

Response 2:

Thank you for the suggested improvements. The main text was modified to the best of our chances and the yet uploaded supplementary material containing scripts and spectra is now implemented according to the precious suggestions

Comment 3

Supply the MATLAB scripts used for shift alignment, NMR quantification and statistical modelling, or deposit them in a public repository (GitHub/Zenodo) with a DOI.
Justify the choice of integrating 19 variables.

Ensure that the file referenced are uploaded and contains at minimum: raw FIDs, processed spectra, code, integrals, climate data for the campaign, etc.

Response 3:

Main text, specifically the last paragraph of 2.6 and supplementary material were implemented according to the important observations. Some of the requested material is not exportable according to our university policy, however all the data reported in the Method section (2) and in the supplementary material allows the possible reproduction of the whole experimental procedure

Comment 4:

A proofreading pass is needed, as form example “secoridoid” should be “secoiridoid” “worthed information”, replace non-standard abbreviation “MUF9” with the accepted lipid shorthand, some journal abbreviations and italics inconsistencies, etc.

Given the scientific interest of the manuscript and the fact that all shortcomings are, in principle, curable by rewriting, re-analysis and improved documentation, I recommend major revision. Please address each comment point-by-point and provide a tracked-changes version of the manuscript.

Response 4:

Typing and editing modifications were done according to the reviewer suggestions.

We thankfully have done our best to fulfil all the reviewer suggestions, several parts of the paper are rewritten and tracked-changes will be visible, several figures and tables are also added to the supplementary material

Reviewer 4 Report

Comments and Suggestions for Authors

The paper “Restored Intensities from Customized Crops of NMR experiments (RICC-NMR) to gain a better insight on chemometrics of Sicilian and Sardinian Extra Virgin Olive Oils” was an interesting paper to be read dealing with that describes an NMR based procedure (a new protocol) named Restored Intensities from Customized Crops of NMR experiments (RICC-NMR). The exemplified application of the entire procedure aimed to find relevant parameters (and statistical analysis) to discriminate between Sicilian (SIC) and Sardinian (SAR) samples belonging to the same Olea europaea strain. You proved that this approach is successful and, in my opinion, the conclusions are sustained by experiments. Despite my numerous observations (see below) these are related to some points that must be clarified but will not question the research procedure and have no impact on the research findings, approach, applications and conclusions. By this manuscript, you provide a procedure (experimental ana analytical/numerical) able to discriminate between closely related samples but with different origin. In my opinion, the value of your procedure is the fact that it can be successfully extrapolated to other types of foods but can also be extended to other research area, such as medicine (e.g. for cancer detection). Of course, a 600-MHz Bruker spectrometer is expensive. Below you have my remarks:
1.    Abstract: It is unclear how many NMR experiments were performed. “four different Nuclear Magnetic Resonance (NMR) experiments”; “The acquisition of three different 1H-NMR experiments was set”; “The robust setup of four experiments, namely I) standard 1H{13C}, 30 II) multiple pre-saturated 1H{13C}; III) 1H selective excitation at 9.25 ppm, and IV) 13C{1H} 31 acquisition, allowed to process the 1H-RICC-NMR and the 13C-NMR profiles”; “1H NMR spectrum was reconstructed by customized crops from the three mentioned experiments (RICC-NMR)”. Please be more specific. 
2.    Abstract: “NMR experiments”, please be more specific. NMR may involve spectroscopy, relaxometry, diffusiometry, imaging, etc. then the experiments can be 1D, 2D or nD (dimensional).
3.    Abstract: the origin of “RICC-NMR” it is unclear. Please be more specific. In title it is clear.
4.    Abstract: The main objective is unclear. It is about NMR? There are some novelties? It is about oils?
5.    Introduction. You mention “In this study, we present an original experimental protocol”. Please describe briefly the previous protocol(s) using technical details and describe the problem (limitation), and how you proposed to improve the acquisition (and analysis) protocol.
6.    Section 2.3 NMR Sample preparation. Please explain briefly the effect of dilution in high-purity deuterated chloroform (CDCl3) on the NMR spectra! Why and how you can increase the sensitivity.
7.    Line 111 (and others): Please clarify the term “cycling delay”. Some standards in NMR use “recycle delay” and “phase cycling”. Are there related? In line 120 “recycling delay” is clear.
8.    Line 134 “The 1D, 1H{13C}. Selective” – a too short sentence.
9.    Line 140 “protonic” -> “proton relaxation time”
10.    Lines 140-141 “maximum value of T1 = 2.5, which is less than 5 times the total recycling time”. Something seems to be wrongly explained here! First please use the measurement units, e.g. “T1 = 2.5 s”, with T italic and 1 – subscript. Then, it is the “5 x total recycling time”? Usually the recycle delay (RD) should be RD > 5 T_1 (the longitudinal or spin-lattice relaxation time). Please clarify these aspects.
11.    Lines 143-144 “The shaped pulse between 8 and 10 ppm was an optimized BURP (Band- Selective, Uniform Response, Pure-Phase) [24] with a duration of 7 ms.” The pulse is in time, and the spectrum is in frequency (or ppm). Then a “pulse between 8 and 10 ppm” is unphysical. Please express clear this idea! It is produced with a frequency (offset) corresponding to that chemical shift?
12.    Lines 144-146 “After 8 to 16 scans, the spectrum displayed signals in the region of interest (barely foreseen through experiments I, and II) with a better signal-to-noise ratio” – please rephrase. I assumed that you meant that the signal-to-noise ratio is good enough after 8 scans so the spectral lines start to be observed.
13.    Line 146-147: “slight line-broadening function (0.3 Hz) was applied for the Fourier-transform procedure” –is this line-broadening function one of the standard one, such as Gaussian or Lorentzian?
14.    Line 150 “relaxation delay (RD),” –unclear. RD usually is recycling delay. It is also true that there the longitudinal relaxation processes (with longitudinal relaxation time T_1) takes places and the sample it is re-magnetized. Please clarify.
15.    Lines 148-150. Please clarify some aspects: i) “amplitude- and phase-modulated shaped pulse was applied during the relaxation delay (RD)” – usually there are no pulses during recycle delays. This is true with the exception of the slice selection. It was your case? Please clarify. ii) “highly selective frequency bands” and “modulated shaped pulse” it is applied in the presence of a gradient of magnetic fields. It was your case?
16.    Line 156 “time delay” – please be consistent with the name and notation of NMR parameters all over the manuscript! (see also the next lines: “recycling time” and “total recycling delay for a total of 48 minutes of experimental time”.).
17.    Line 186 “The integration of 102 regions of RICC-NMR profile and 98 regions from the 13C-NMR enabled”. Please provide information related to how do you chose these 102 regions for RICC-NMR profile and 98 regions from the 13C-NMR? Equidistant or not? In which range? Continuous or sparse?
18.    Line 187 “quantification of 19 chemical compounds” – are these presented in Table 1? If yes please state that in this line! Also please add the number in Table 1 as the firs column.
19.    Line 201 “Traditional analytical essays”; “essays” or “assays”?
20.    Lines 206-209. Please correlate “the health conditions of the raw material” with the “percentage of oleic acid”.
21.    Line 212 what are the “K232, K270” parameters?
22.    Line 243 and figure 2. Please explain (here) the SIC and SAR acronyms! 
23.    Many of the information (all non-specific) provided in the Discussion section must be part of the introduction, since the aim, the problem and the solution should be presented to the readers not in the final, but from the beginning. 
24.    The association of the quantified compounds with the position in the RICC-NMR spectrum will be welcomed. In this sense “The use of different spectra enabled the extraction of targeted extensive information on minor components through definite integrals, which were typically obscured by more intense signals within standard NMR experiments.” It will better sustained by some examples of the quantified compounds and their position in the RICC-NMR spectrum.

Author Response

We are very thankful to the reviewer for his/her suggestions and indications which hopefully would improve the quality of the paper. We list below our best response (in red) to all the formulated comments.

Regards

Comment 1:

The paper “Restored Intensities from Customized Crops of NMR experiments (RICC-NMR) to gain a better insight on chemometrics of Sicilian and Sardinian Extra Virgin Olive Oils” was an interesting paper to be read dealing with that describes an NMR based procedure (a new protocol) named Restored Intensities from Customized Crops of NMR experiments (RICC-NMR). The exemplified application of the entire procedure aimed to find relevant parameters (and statistical analysis) to discriminate between Sicilian (SIC) and Sardinian (SAR) samples belonging to the same Olea europaea strain. You proved that this approach is sucessful and, in my opinion, the conclusions are sustained by experiments. Despite my numerous observations (see below) these are related to some points that must be clarified but will not question the research procedure and have no impact on the research findings, approach, applications and conclusions. By this manuscript, you provide a procedure (experimental ana analytical/numerical) able to discriminate between closely related samples but with different origin. In my opinion, the value of your procedure is the fact that it can be successfully extrapolated to other types of foods but can also be extended to other research area, such as medicine (e.g. for cancer detection). Of course, a 600-MHz Bruker spectrometer is expensive.

Response 1:

We thank the reviewer for the very clever general comment which is kept into consideration (see conclusions) also in the massive editing suggested by the other reviewers. We totally appreciate and agree these general statements.

Comment 2:

Below you have my remarks:
1.    Abstract: It is unclear how many NMR experiments were performed. “four different Nuclear Magnetic Resonance (NMR) experiments”; “The acquisition of three different 1H-NMR experiments was set”; “The robust setup of four experiments, namely I) standard 1H{13C}, 30 II) multiple pre-saturated 1H{13C}; III) 1H selective excitation at 9.25 ppm, and IV) 13C{1H} 31 acquisition, allowed to process the 1H-RICC-NMR and the 13C-NMR profiles”; “1H NMR spectrum was reconstructed by customized crops from the three mentioned experiments (RICC-NMR)”. Please be more specific. 
2.    Abstract: “NMR experiments”, please be more specific. NMR may involve spectroscopy, relaxometry, diffusiometry, imaging, etc. then the experiments can be 1D, 2D or nD (dimensional).
3.    Abstract: the origin of “RICC-NMR” it is unclear. Please be more specific. In title it is clear.
4.    Abstract: The main objective is unclear. It is about NMR? There are some novelties? It is about oils?

Response 2:

We express once again our gratitude for pointing out the unclear abstract which was totally rewritten by respecting the reported indications

Comments 3:

  1.    Introduction. You mention “In this study, we present an original experimental protocol”. Please describe briefly the previous protocol(s) using technical details and describe the problem (limitation), and how you proposed to improve the acquisition (and analysis) protocol.

Response 3:

We have modified the final paragraph of the introduction accordingly and mentioned references describe the previous protocols.

Comment 4:
6.    Section 2.3 NMR Sample preparation. Please explain briefly the effect of dilution in high-purity deuterated chloroform (CDCl3) on the NMR spectra! Why and how you can increase the sensitivity.
7.    Line 111 (and others): Please clarify the term “cycling delay”. Some standards in NMR use “recycle delay” and “phase cycling”. Are there related? In line 120 “recycling delay” is clear.
8.    Line 134 “The 1D, 1H{13C}. Selective” – a too short sentence.
9.    Line 140 “protonic” -> “proton relaxation time”
10.    Lines 140-141 “maximum value of T1 = 2.5, which is less than 5 times the total recycling time”. Something seems to be wrongly explained here! First please use the measurement units, e.g. “T1 = 2.5 s”, with T italic and 1 – subscript. Then, it is the “5 x total recycling time”? Usually the recycle delay (RD) should be RD > 5 T_1 (the longitudinal or spin-lattice relaxation time). Please clarify these aspects.
11.    Lines 143-144 “The shaped pulse between 8 and 10 ppm was an optimized BURP (Band- Selective, Uniform Response, Pure-Phase) [24] with a duration of 7 ms.” The pulse is in time, and the spectrum is in frequency (or ppm). Then a “pulse between 8 and 10 ppm” is unphysical. Please express clear this idea! It is produced with a frequency (offset) corresponding to that chemical shift?
12.    Lines 144-146 “After 8 to 16 scans, the spectrum displayed signals in the region of interest (barely foreseen through experiments I, and II) with a better signal-to-noise ratio” – please rephrase. I assumed that you meant that the signal-to-noise ratio is good enough after 8 scans so the spectral lines start to be observed.
13.    Line 146-147: “slight line-broadening function (0.3 Hz) was applied for the Fourier-transform procedure” –is this line-broadening function one of the standard one, such as Gaussian or Lorentzian?

  1.    Line 150 “relaxation delay (RD),” –unclear. RD usually is recycling delay. It is also true that there the longitudinal relaxation processes (with longitudinal relaxation time T_1) takes places and the sample it is re-magnetized. Please clarify.

Response 4:

Thank you, We admit several theoretical, conceptual and typing flaws which are now fixed to our best.

Comment 5:
15.    Lines 148-150. Please clarify some aspects: i) “amplitude- and phase-modulated shaped pulse was applied during the relaxation delay (RD)” – usually there are no pulses during recycle delays. This is true with the exception of the slice selection. It was your case? Please clarify. ii) “highly selective frequency bands” and “modulated shaped pulse” it is applied in the presence of a gradient of magnetic fields. It was your case?
16.    Line 156 “time delay” – please be consistent with the name and notation of NMR parameters all over the manuscript! (see also the next lines: “recycling time” and “total recycling delay for a total of 48 minutes of experimental time”.).

Response 5:

Thank you, We admit several theoretical, conceptual and typing flaws which are now fixed to our best.

Comment 6
17.    Line 186 “The integration of 102 regions of RICC-NMR profile and 98 regions from the 13C-NMR enabled”. Please provide information related to how do you chose these 102 regions for RICC-NMR profile and 98 regions from the 13C-NMR? Equidistant or not? In which range? Continuous or sparse?
18.    Line 187 “quantification of 19 chemical compounds” – are these presented in Table 1? If yes please state that in this line! Also please add the number in Table 1 as the firs column.

Response 6:

Thanks to your observations we have widely implemented the section 2.6 and correspondingly the supplementary material to clarify the risen doubts.

Comment 7:

  1.    Line 201 “Traditional analytical essays”; “essays” or “assays”?
    20.    Lines 206-209. Please correlate “the health conditions of the raw material” with the “percentage of oleic acid”.
    21.    Line 212 what are the “K232, K270” parameters?

Response 7:

According to other reviewers we have deleted the section 2.7

Comment 8:

  1.    Line 243 and figure 2. Please explain (here) the SIC and SAR acronyms! 
    23.    Many of the information (all non-specific) provided in the Discussion section must be part of the introduction, since the aim, the problem and the solution should be presented to the readers not in the final, but from the beginning. 
    24.    The association of the quantified compounds with the position in the RICC-NMR spectrum will be welcomed. In this sense “The use of different spectra enabled the extraction of targeted extensive information on minor components through definite integrals, which were typically obscured by more intense signals within standard NMR experiments.” It will better sustained by some examples of the quantified compounds and their position in the RICC-NMR spectrum.

Response 8:

We have tried our best to clarify all these last appropriate observations by modifying the main tex along with the integrated supplementary material